# Development of PLGA-PEG-PLGA Hydrogel Delivery System for Enhanced Immunoreaction and Efficacy of Newcastle Disease Virus DNA Vaccine

**DOI:** 10.3390/molecules25112505

**Published:** 2020-05-28

**Authors:** Ying Gao, Hui Ji, Lin Peng, Xiuge Gao, Shanxiang Jiang

**Affiliations:** Laboratory of Veterinary Pharmacology and Toxicology, College of Veterinary Medicine, Nanjing Agricultural University, Nanjing 210095, China; gaoyeen@outlook.com (Y.G.); jihui1019@163.com (H.J.); pl18951630156@163.com (L.P.); vetgao@njau.edu.cn (X.G.)

**Keywords:** Newcastle disease, DNA vaccine, delivery system, PLGA-PEG-PLGA, hydrogel

## Abstract

The highly contagious Newcastle disease virus (NDV) continues to threaten poultry all over the world. The NDV DNA vaccine is a promising solution to the current Newcastle disease (ND) challenges, and thus an efficient delivery system should be developed to facilitate the efficacy of DNA vaccines. In this study, we developed a DNA vaccine delivery system consisting of a triblock copolymer of poly(lactide co-glycolide acid) and polyethylene glycol (PLGA-PEG-PLGA) hydrogel in which the recombinant NDV hemagglutinin-neuraminidase (HN) plasmid was encapsulated. Its characteristics, security, immune responses, and efficacy against highly virulent NDV were detected. The results showed that the plasmids were gradually released in a sustained manner from the hydrogel, which improved the biological stability of the plasmids and demonstrated a high biocompatibility. The plasmids, when they were incorporated into the hydrogel delivery system, enhanced immune activation and provided 100% protection against the highly virulent NDV strain. Furthermore, we proved that this NDV DNA hydrogel vaccine could improve the lymphocyte proliferation and increase the immunological cytokine production via the PI3K/Akt pathway. These results indicate that the PLGA-PEG-PLGA thermosensitive hydrogel could be a promising delivery system for the NDV DNA vaccine in order to achieve a sustained supply of plasmids and induce potent immune responses.

## 1. Introduction

Newcastle disease (ND) is a highly contagious viral disease in poultry which leads to high mortality and huge economic losses in the poultry industry. Therefore, ND was confirmed in the list of diseases requiring urgent attention by the World Organization for Animal Health [1]. The Newcastle disease virus (NDV) is an avian paramyxovirus type 1 virus belonging to the genus Avulavirus within the family Paramyxoviridae [2]. The genome of NDV is a non-segmented, negative-sense, single-stranded RNA genome of 15,186 to 15,198 nucleotides in length, encoding six structural proteins: nuclear protein (N), phosphoprotein (P), large polymerase protein (L), matrix protein (M), hemagglutinin-neuraminidase (HN), and fusion protein (F). Among these proteins, the HN protein, as the main protective antigen on the surface of NDV, has good immunogenicity [3,4]. 

There are currently no suitable therapies for ND; however, preventing NDV by vaccination seems to be an effective method. Although attenuated live and inactivated vaccines are utilized universally, ND is still a devastating disease in many countries [5]. The conventional ND vaccines have some shortcomings. For example, the inactivated vaccines generally have poor immunogenicity and may lead to death due to incomplete inactivation [6]. In addition, attenuated live vaccines are limited by some factors, such as virulence reversion and high requirements for cold chain and biocontainment [7]. Hence, it is necessary to develop novel ND vaccines to improve safety and efficiency based on the current situation.

Advances in recombinant DNA technology have made a DNA vaccine a promising solution to current ND challenges. When the recombinant expression plasmids that encode one or more immunogens are directly administered to the host, the cloned genes would be expressed in vivo and translated into proteins, serving as potent antigens capable of inducing a protective immune response. The major merits of the DNA vaccine are its safety, convenience, and ability to activate innate immunity responses. The DNA vaccine also has its limitations, such as the low level of target gene expression, low bioavailability, degradation before reaching antigen presenting cells, and, thus, weak immune response [8,9]. An approach that can solve these problems is thus desired.

Poly(lactide co-glycolide acid) and polyethylene glycol triblock copolymer (PLGA-PEG-PLGA) is a biodegradable thermosensitive hydrogel that appears to be an attractive option for the vaccine delivery system on account of its beneficial properties, such as good biocompatibility, controllable biodegradation rate, adjustable sol-gel transition, easy administration, and effective drug delivery. This triblock copolymer, which contains both hydrophilic and hydrophobic parts, exhibits thermosensitive properties. It stays in a liquid state at room temperature, which facilitates mixing with recombinant plasmids and subsequent injection. When inoculated into the host, the polymer will achieve a sol-gel transition at body temperature. Additionally, many conventional hydrogels can only release drugs for a few days [10]. However, the PLGA-PEG-PLGA hydrogel can constantly release the incorporated drugs over weeks or months at the injection site [11]. Therefore, PLGA-PEG-PLGA hydrogel is expected to be a promising vaccine delivery system that can be simply prepared by mixing and is capable of sustained antigen release.

Based on these considerations, this study aimed to develop an appropriate DNA vaccine delivery system that could continuously release plasmids and protect them from degradation, improving the biological stability and inducing strong immunostimulatory activities. In this case, we developed a kind of PLGA-PEG-PLGA thermosensitive hydrogel and tested its characteristics and biocompatibility. Then, the recombinant plasmids of the NDV HN gene (pVAX1-HN) were prepared and encapsulated into the hydrogel to obtain the NDV DNA hydrogel vaccine, or Gel-pVAX1-HN. We first evaluated the plasmid release from the hydrogel and its biological stability. We then examined the immune responses and the protection against NDV after the inoculation of the chickens. Finally, we investigated the effects of Gel-pVAX1-HN on the proliferation and activation of lymphocytes for disclosing the mechanism of the immune enhancement induced by the vaccine delivery system. 

## 2. Results

### 2.1. Characterization of PLGA-PEG-PLGA Hydrogel

Figure 1a shows the 1H-NMR spectrum of the PLGA-PEG-PLGA copolymer. The characteristic signals appearing at 5.2, 4.8, 3.6, and 1.5 ppm are assigned to the methine hydrogen of the DL-lactide units, the methylene hydrogen of the polyethylene glycol (PEG), and the methyl hydrogen of the DL-lactide units, respectively. The DL-lactide/glycolide molar ratio is 3.03/1. Table 1 shows the chemical shifts and corresponding protons of the PLGA-PEG-PLGA polymer.

The phase-transition temperature (Tm), gelation time (Gt), and viscosity of the aqueous solutions of the PLGA-PEG-PLGA copolymer at various concentrations were determined to elect the optimal concentration for the copolymer-based hydrogel. As shown in Figure 1b, when increasing the copolymer concentrations from 15% (*w*/*w*) to 22% (*w*/*w*), the Tm and Gt gradually decreased and the viscosity increased, respectively. Based on the suitable Tm (35.3 °C) and short Gt (64.7 s), the concentration at 20% was used for the following experiments.

The polymer solution at a concentration of 20% was transparent liquid at 25 °C, while it presented a semi-solid gel state when the temperature was higher than 35.3 °C (Figure 1c). The micelles of the polymer solution observed by transmission electron microscopy (TEM) were of a regular spherical morphology with smooth surfaces, dispersed well without aggregation, and possessed a uniform particle size (Figure 1d). The average size of the micelles calculated by dynamic light scattering (DLS) analyses was 26.02 nm, and the polydispersity index (PDI) was 0.249. In addition, the particle size of micelles with different dilution ratios remained constant (Figure 1e). The zeta potential of micelles was −0.16 mV, indicating that the copolymer was electrical neutrality (Figure 1f). 

### 2.2. Safety of PLGA-PEG-PLGA Hydrogel

Safety is a fundamental prerequisite for biomaterials used in vaccine delivery. Here, the in vitro and in vivo experiments were carried out to examine the biocompatibility of the PLGA-PEG-PLGA hydrogel. A hemolysis test is a common physiological acute toxicity response that ensures the biocompatibility of injectable biomaterials. We added 20% copolymer solution into the chicken blood at five different volumes and incubated at 37 °C for 3 h. The results displayed that the copolymer solution did not cause hemolysis (Figure 2a). To measure the cytotoxicity of the copolymer, chicken embryonic fibroblasts and HD11 (chicken macrophage cell line) were seeded in the 96-well plates and cultured with 10 μL of 20% polymer or phosphate buffer saline (PBS) for 48 h. The CCK-8 assay results demonstrated that the hydrogel was non-toxic to the cells (Figure 2b). Then, we carried out the in vivo safety test. After subcutaneous and intramuscular injections with 1ml hydrogel, no nervous signs, clinical symptoms, or necropsy lesions were observed. After 14 days, all the chickens survived, and the skin and muscle samples from the injection site were collected for hematoxylin-eosin (HE) staining; the results revealed that the tissue structures were normal, and no inflammatory cell infiltration was observed (Figure 2c).

### 2.3. Plasmid Release from PLGA-PEG-PLGA Hydrogel 

To evaluate the sustained release property of the hydrogels for the DNA vaccine, we constructed the HN gene of NDV into the pVAX1 plasmid then enveloped 0.5 mg·mL^−1^ of the recombined plasmid (pVAX1-HN) in the hydrogel; the release profile was evaluated using the membrane-free dissolution method. As shown in Figure 3a, the PLGA-PEG-PLGA hydrogel could continuously release plasmids for 22 days, with a cumulative release rate of 95.07% and an effective drug loading of 0.47 mg·g^−1^. The release media of pVAX1-HN from the hydrogel were collected at different times. The electrophoresis image showed that the plasmid structure was intact and not degraded (Figure 3b). When the release media were transfected into the HD11 cells, the immune fluorescence results revealed that the biological activity of the plasmids remained constant during the release process (Figure 3c).

### 2.4. PLGA-PEG-PLGA Hydrogel Enhanced DNA Vaccine-Induced Humoral Immunity

In order to evaluate the immune effects of the NDV gel vaccine, some chickens aged 14 days were inoculated with the vaccine by intramuscular injection and serum samples were collected at each predetermined time point after immunization. The HI titers and concentrations of the NDV antibody (NDV-Ab) were determined. The antibody titers in the pVAX1-HN group (naked HN plasmids without hydrogel) slightly increased after the inoculation and soon decreased. In the LaSota group (commercial attenuated vaccine), the serum HI titers initially increased and then declined. In the Gel-pVAX1-HN group, the HI titers increased during the first week after immunization and steadily rose to the maximum of 7.0 log2; they were maintained at a high level (greater than 6.0 log2) during the sixth week. Meanwhile, the concentrations of the NDV antibody peaked (1686 ± 54 ng·ml^−1^) at the third week after immunization and remained at high levels until the end of monitoring; they were significantly higher than the levels of the naked plasmid group (*P* < 0.05) (Figure 4b). Thus, the PLGA-PEG-PLGA hydrogel could significantly boost the DNA vaccine-induced humoral immunity.

### 2.5. PLGA-PEG-PLGA Hydrogel Enhanced DNA Vaccine-Induced Cellular Immunity 

The concentrations of IL-2, IL-4, and IFN-γ were determined in serum collected at different times after immunization. Meanwhile, the peripheral blood lymphocytes were isolated and cultured to determine the proliferation activity. As shown in Figure 5a–c, the secretion of IL-2, IL-4, and IFN-γ was significantly promoted and maintained at high levels for a long time. The proliferation activity of lymphocytes was significantly higher than that of the naked plasmid group as well as the commercial vaccine group (*p* < 0.05). To further measure the cellular immune levels of the immunized chickens, the percentages of CD4^+^ and CD8^+^ T cells in the peripheral blood lymphocytes were determined. The flow cytometry results showed that the percentage of CD4^+^ and CD8^+^ T cells in the Gel-pVAX1-HN group at the third week were significantly higher than in the other groups (Figure 6a,b). These results suggest that the NDV DNA hydrogel vaccine induced a strong cellular immunity and remained at a high level for a period of time.

### 2.6. PLGA-PEG-PLGA Hydrogel Enhanced Protective Efficacy of NDV DNA Vaccine

At the end of the monitoring period, the chickens were challenged with the virulent strain of NDV to detect the protection rates of the gel vaccine. Fourteen days later, there were no clinical symptoms and no mortality in the chickens immunized with the Gel-pVAX1-HN and LaSota vaccines; the protection rates against the virulent strain of NDV were 100%, while the feeding, drinking, and mental states were at normal levels. In the pVAX1-HN group, the protective efficacy was 60%. Meanwhile, all the chickens in the PBS and pVAX1 groups exhibited typical NDV-infected signs then died within 4 to 7 days (Figure 7). These findings indicated that the Gel-pVAX1-HN vaccine significantly enhanced immunity function and provided a good protection efficacy against NDV.

### 2.7. PLGA-PEG-PLGA Hydrogel Enhanced the Activation of PI3K/Akt Pathway

The above findings demonstrated that the Gel-pVAX1-HN vaccine induced lymphocyte proliferation and immune-cytokine production. The PI3K/Akt signaling pathway plays a vital role in cell proliferation and inflammatory generation. Hence, we analyzed the protein expressions in the peripheral blood lymphocytes. The western blot results displayed that the phosphorylation levels of PDK1, Akt, and IkB-α were significantly elevated in lymphocytes from the Gel-pVAX1-HN group, suggesting that the Gel-pVAX1-HN vaccine activated the PI3K/Akt-signaling pathway (Figure 8a,b). Upon treating lymphocytes with LY294002 (PI3K inhibitor), the Gel-pVAX1-HN-induced lymphocyte proliferation and cytokine production were both suppressed (Figure 8c,d). These results proved that the immune enhancement effects of the NDV gel vaccine were dependent on the activation of the PI3K/Akt pathway in the lymphocytes.

## 3. Discussion

Since 1926, ND has accounted for tremendous economic losses to the poultry industry. Because of the high infectivity of NDV and the drawbacks of conventional vaccines, ND is still one of the most severe avian diseases all over the world [12]. It is imperative to develop more efficient vaccines and relative vaccine delivery systems to resist ND challenges across the globe. The DNA vaccine presents several advantages over conventional vaccines, such as its security and ability to be a multivalent vaccine. It also can induce both humoral and cellular immunity, which are more effective in preventing diseases such as viral infections that rely on cellular immune clearance [8]. However, the DNA vaccine requires multiple and large doses of vaccination to induce effective immune responses. In addition, the expression plasmids are easily affected by pH, enzymes, and other factors in vivo and are degraded and inactivated, which limits the clinical applications of DNA vaccine [9]. To enhance the effectiveness of the DNA vaccine, the sustained stimulation of the immune system and protection of the antigen from degradation is desirable. PLGA-PEG-PLGA polymer has been used as a peptide and protein drug delivery system for decades. However, the studies on its applications as a DNA vaccine delivery system required more attention. Hence, we prepared the PLGA-PEG-PLGA hydrogel as carrier for the NDV DNA vaccine to improve its protective efficacy against NDV.

The T_m_ and Gt of thermosensitive hydrogel are vital factors in the vaccine delivery system. In this study, PLGA-PEG-PLGA polymer was synthesized by ring-opening polymerization and the optimal concentration of the purified polymer was determined to be 20% through prescription optimization. The suitable T_m_ (35.3 °C) and relatively short Gt (64.7 s) indicated stability at room temperature and a rapid sol-gel conversion in vivo. The particle sizes of the polymer micelles are uniform, with an average diameter of 26.02 nm, revealing the micellar homogeneity of the hydrogel. In addition, zeta potential is a key indicator of polymer safety. Polymers with excessive potential have a strong cytotoxicity, which is caused by the accumulation of too many charges on the cell membrane surface, resulting in the rupture of the cells [13]. According to the latest research, the zeta potential of PLGA-PEG-PLGA aqueous solution is close to electrical neutrality [14], which is consistent with our findings (−0.16 mV).

Security is the precondition for the application of biomaterials in vivo. Hemolysis test results revealed that the PLGA-PEG-PLGA polymer would not cause hemolysis and erythrocyte coagulation. Macrophages play an essential role in eliminating pathogens, serving as the first line of defense and mediating pivotal innate immune responses [15]. The chicken macrophage cell line (HD11) and embryo fibroblasts were used for a cytotoxicity test in vitro, and it turned out that the triblock copolymer had no significant effect on cell survival rates. An in vivo cytotoxicity evaluation through subcutaneous and intramuscular injections showed that the copolymer had no obvious damage to the surrounding tissues of the injection sites. Therefore, the results of the safety evaluation indicated that the PLGA-PEG-PLGA polymer has a high biocompatibility for follow-up studies of the vaccine delivery system. 

The antibody detection results proved that the hydrogel vaccine upregulated the humoral immunity in vivo. Without hydrogel, the naked plasmids only induced a limited humoral immune response and soon declined. When the log2 fold change of the antibody titer is lower than 4, it will not provide effective protection against NDV [1]. It is worth noting that the plasmids encapsulated in the PLGA-PEG-PLGA hydrogel not only enhanced the humoral immunity but also promoted cellular immunity. The cellular immune responses were assessed by detecting the lymphocyte proliferation and immune cytokine production. The proliferation of lymphocytes, as vital immune cells, is a critical event in the activation of immune responses [16]. In the Gel-pVAX1-HN group, the proliferation activity of lymphocytes was significantly higher than that of the naked plasmid group as well as the commercial vaccine group. IL-2, IL-4, and IFN-γ are pivotal cytokines mediating immune responses [17,18]. In this study, the secretion of the three cytokines was significantly promoted and maintained at high levels for a long time. CD4^+^ T helper cells and CD8^+^ cytotoxic T cells are two essential components of T lymphocytes that play key roles in immune responses [19]. It is not surprising that the ratios of CD4^+^ and CD8^+^ T cells increased significantly in Gel-pVAX1-HN group. These results implied that the DNA hydrogel vaccine promoted both humoral and cellular immune responses. The high immunostimulatory activity induced by the NDV hydrogel vaccine is considered a result of the improved stabilization and sustained release of plasmids in vivo [20].

Studies have shown that the PI3K/Akt pathway plays dominated roles in the cell proliferation and generation of immunological cytokines [21]. Therefore, it was presumed that the PI3K/Akt pathway might contribute to the immune enhancement inducted by the NDV hydrogel vaccine. In view of this, the protein expression of the PI3K/Akt pathway in lymphocytes after immunization was examined. The western blot results revealed that the PI3K/Akt pathway in lymphocytes was obviously activated, inducing significant lymphocyte proliferation and cytokine production in the Gel-pVAX1-HN group, whereas the enhancement effects were significantly eliminated when using the PI3K inhibitor. Our study suggests that the immune enhancement effects of the NDV hydrogel vaccine were dependent on the activation of the PI3K/Akt pathway in lymphocytes.

In conclusion, this study revealed an approach that can solve the problems associated with DNA vaccines. The DNA hydrogel vaccine developed in this study was composed only of eukaryotic expression plasmids and PLGA-PEG-PLGA hydrogel and was easy to prepare by mixing. The plasmids could be efficiently loaded into the hydrogel and released slowly with improved biological stability. This vaccine exhibited immune enhancement effects through accelerating the lymphocyte proliferation and cytokine production regulated by the PI3K-Akt pathway activation. Moreover, the protective efficacy of the DNA hydrogel vaccine against highly virulent NDV reached 100%. The results of the present study indicate that the PLGA-PEG-PLGA thermosensitive hydrogel could be used as a promising DNA vaccine delivery system to induce potent immunoreactions and increase the vaccine potency.

## 4. Materials and Methods 

### 4.1. Materials and Animals

Polyethylene glycol (PEG1500) and stannous 2-ethylhexanoate were purchased form Sigma (Santa Clara, CA, USA). DL-lactide and glycolide were purchased from Sinopharm Chemical Co. Ltd. and used without further purification (Nanjing, China). The recombinant plasmids that express the HN gene of NDV (pVAX1-HN) were supplied by the Laboratory of Veterinary Pharmacology and Toxicology, Nanjing Agricultural University (Nanjing, China). The commercial NDV-attenuated vaccine (LaSota strain) was purchased from the YEBIO Corporation (Qingdao, China). The highly virulent NDV (F48E9 strain) and NDV-positive serum were obtained by Jiangsu Academy of Agricultural Sciences (Nanjing, China). One-day-old Hy-line white chickens (male), provided from Tangquan Poultry Farm (Nanjing, China), were housed in wire cages (60 cm × 100 cm) in air-conditioned rooms at 37 °C and lighted for 24 h at the beginning of the pretrial period. The temperature was gradually adjusted to the room temperature and the light time to 12 h per day, and kept constant in the following days. All of the experiments related to animals were performed in accordance with the guidelines of the regional Animal Ethics Committee and were approved by the Institutional Animal Care and Use Committee of Nanjing Agricultural University (Nanjing, China) (NJAU-CAST-2018-179). 

### 4.2. Synthesis of PLGA-PEG-PLGA Triblock Copolymer

DL-lactide, glycolide, PEG1500, and stannous 2-ethylhexanoate (0.2%, *w*/*w*) were accurately weighed and added into a dried reaction flask of 50 mL. Then, the flask was put into a freeze-dryer for vacuum extraction and sealed under vacuum. The sealed flask was immersed and kept in an oil bath heated at 140 °C for 12 h, and the colorless and transparent semi-solid product was obtained. Then, the product was completely dissolved in purified water of 4 °C and heated to 70 °C. The copolymer was precipitated and the water-soluble low-molecular weight copolymer and unreacted monomers in the water layer were discarded. The above purification process was repeated three times to obtain the purified PLGA-PEG-PLGA triblock copolymer, which was a light yellow sticky paste. The ^1^H nuclear magnetic resonance (NMR) spectra of the PLGA-PEG-PLGA copolymer were obtained in CDCl_3_ using a NMR instrument (Bruker ARX-300, Bruker Corporation, Billerica, MA, USA) at 300 MHz. The DL-lactide to glycolide mole ratio of the copolymer was calculated according to the ^1^H NMR result.

### 4.3. Measurement of T_m_, Gt and Viscosity 

The PLGA-PEG-PLGA copolymers were precisely measured and dissolved in distilled water. The polymers were evenly dispersed by proper stirring and placed at 4 °C until completely dissolved to obtain homogeneous hydrogel solutions with different concentrations (15%–22%). Then, the solution was placed into a glass tube and heated at a constant rate of 1 °C per minute in a water bath. When the solution was observed to be stagnant with the tilt of the tube and this lasted for more than 30 s, it was considered that the solution had been converted to hydrogel. The temperature read from the thermometer was determined as the phase-transition temperature (T_m_), and this period was defined as the gelation time (Gt). The viscosity of the hydrogel solutions with different concentrations at room temperature was detected by a rotary viscometer.

### 4.4. Hemolysis Test and Cytotoxicity Assay

A hemolysis test was carried out, referring to the method described in previous studies [22]. Healthy chicken’s blood was collected into anticoagulant tubes and subsequently diluted into a 1% suspension with PBS. An amount of 2.5 mL of blood suspension was taken into tubes, and the copolymer solutions at 0.5, 0.4, 0.3, 0.2, 0.1, 0 mL (n = 3) were added. Each tube was filled to 5 mL with distilled water and incubated at 37 °C for 3 h. Then, the tubes were centrifuged at 1500 rpm for 10 min at 4 °C, the supernatants were assessed for the visible absorbance spectra in 540 nm by using a spectrophotometer, and the hemolysis ratios were calculated using the equation as follows:(1)$$\mathrm{Hemolysis}\;\mathrm{ratio}\;(\%)=\frac{\mathrm{OD}540\;\mathrm{of}\;\mathrm{treatment}\;\mathrm{group}-\mathrm{OD}540\;\mathrm{of}\;\mathrm{negative}\;\mathrm{group}}{\mathrm{OD}540\;\mathrm{of}\;\mathrm{positive}\;\mathrm{group}-\mathrm{OD}540\;\mathrm{of}\;\mathrm{negative}\;\mathrm{group}}\;\times 100\%.$$

For the determination of the cytotoxicity of the polymer, the chicken embryonic fibroblasts and HD11 cells seeded on 96-well plates at a density of 5 × 10^4^ cells/well were cultured in the Dulbecco’s Modified Eagle Medium (DMEM) with 10% of the fetal bovine serum (FBS). After culturing the cells on the plates overnight, the polymer solution was added and co-cultured for 24 h, 48h, or 72 h at 37 °C. Then, the absorbance was detected, followed by the addition of 10 μL of CCK-8 solution. The results were expressed as percentages of the controls, which were arbitrarily assigned with 100% viability.

### 4.5. Plasmid Release from PLGA-PEG-PLGA Hydrogel

The recombinant plasmids pVAX1-HN were diluted with appropriate amounts of distilled water and slowly added into the same volume of PLGA-PEG-PLGA hydrogel solution. The mixture was fully mixed and placed at 4 °C to obtain a NDV DNA hydrogel vaccine, in which the concentration of the copolymer was 20% (*w*/*w*) and the plasmid concentration was 0.5 mg·ml^−1^. Then, the in vitro release of plasmids from the hydrogel was investigated by the membrane-free dissolution method [23]. Briefly speaking, the DNA hydrogel vaccine of 2 mL was precisely measured, placed into a glass tube and incubated at 41 °C (the temperature of the chicken is 40–42 °C) for 10 min to complete gelation. After that, 2 mL of phosphate buffer (pH 7.4) was added to the tube as a release medium and the tube was shaken at 50 rpm. The release medium was collected at the sampling times and replaced with the same volume of fresh buffer in order to maintain the sink conditions. The obtained release medium was filtered by 0.22 μm microporous filter. The amount of released plasmid was measured, the percentage of cumulative release was calculated, and the cumulative release curve was obtained.

The release medium samples from the DNA hydrogel vaccine were collected at 2, 10, and 18 days, then transfected into HD11 cells with a LipoFiter liposome transfection reagent (Hanbio, Shanghai, China). After 2 days, the cells were incubated with NDV-positive serum and stained with the fluorescein isothiocyanate labelled anti-chicken immunoglobulin G antibody (IgG-FITC) (Sigma, Santa Clara, CA, USA), then observed and photographed by the confocal microscope. 

### 4.6. Immunization and Challenge to Chickens

At day 0, the chickens of 14 days old were randomly assigned to 5 groups (n = 20). Group 1 to 4 was intramuscularly injected twice at a two-week interval with PBS, empty vector plasmids (pVAX1, 200 ng), naked pVAX1-HN plasmids (pVAX1-HN, 200 ng), and commercial NDV-attenuated vaccine (LaSota, 0.05 mL). Group 5 was inoculated only once with the NDV DNA hydrogel vaccine containing 200 ng pVAX1-HN plasmids encapsulated in hydrogel (Gel-pVAX1-HN). Six weeks after the first vaccination, ten chickens from each group were selected randomly and challenged by a nasal drip at a dose of 10^5^ EID_50_ of highly virulent NDV (F48E9 strain). The chickens were monitored for clinical symptoms and mortality for 14 days.

### 4.7. Hemagglutination Inhibition (HI) and Enzyme-linked Immunosorbent Assay (ELISA)

Blood samples from the immunized chickens were clotted at 37 °C and centrifuged at 3000 rpm for 10 min to collect serum. Then, a two-fold serial dilution of serum was prepared in 96-well plates. Next, to each well was added equal volumes (25 μL) of 4 hemagglutination units (4 HAU) NDV antigen, except the last row. Thirty minutes later, 25 μL of 1% chicken erythrocytes were added and incubated for 30 min. The agglutination was monitored and the HI titers were determined as the highest serum dilution completely inhibiting NDV agglutination [1]. Commercial ELISA kits for chickens of NDV-Ab, IL-2, IL-4, and IFN-γ (Senbeijia, Nanjing, China) were used to determine the levels of NDV antibody and cytokine production.

### 4.8. Lymphocyte Proliferation Assay and Flow Cytometry 

Lymphocytes were isolated from the peripheral blood of immunized chickens using commercial kits (Solarbio, Beijing, China) according to the manufacturer’s instructions. Then, the lymphocytes were seeded and cultured in a culture medium with 20 μL of ConA. The plates were cultured in an incubator with 5% CO_2_ for 44 h. The proliferation ratio was determined by the CCK-8 Cell Proliferation and Cytotoxicity Assay Kit (Solarbio, Beijing, China). The cells were treated with 10 μL of CCK-8 solution and incubated at 37 °C for 4 h. The absorbance was measured by using an automated microplate reader (Bio-Rad, Japan) at 450 nm, and the stimulation index (SI) was calculated using the equation as follows: (2)$$\mathrm{SI}=\frac{\mathrm{OD}450\;\mathrm{of}\;\mathrm{stimulated}\;\mathrm{cells}}{\mathrm{OD}450\;\mathrm{of}\;\mathrm{unstimulated}\;\mathrm{cells}}\;\times 100\%.$$

Lymphocytes isolated from different immune chickens were counted and diluted to 1 × 10^6^ cells/mL. The cells of 500 μL were stained with 5 μL of anti-CD4-PE or 4 μL of anti-CD8-PE (ebioscience, San Diego, CA, USA). After 30 min, the cells were washed and examined by flow cytometry.

### 4.9. Western Blot

The total protein extracts with the RIPA (Radio immunoprecipitation assay) lysis buffer and protein quantities were measured by the BCA Protein Assay kit (Auragene Bioscience, Changsha, China). Antibodies against PDK1 (NB100, Novus, Littleton, CO, USA), p-PDK1 (9634S, CST, Boston, MA, USA), Akt (9272S, CST, Boston, MA, USA), p-Akt (104A282, Novus, Littleton, CO, USA), IkBα (ABIN926864, 4A Biotech, Beijing, China), p-IkBα (2859, CST, Boston, MA, USA), and β-actin (ab6276, Abcam, Cambridge, MA, USA) were used in the western blot analysis. Images were captured with the VersaDoc 4000MP system (Bio-Rad).

### 4.10. Statistical Analysis

The data were statistically evaluated by a one-way ANOVA using SPSS 22.0 for multiple comparisons. A *p* value of less than 0.05 was considered statistically significant.

## Figures and Tables

**Figure 1 molecules-25-02505-f001:**
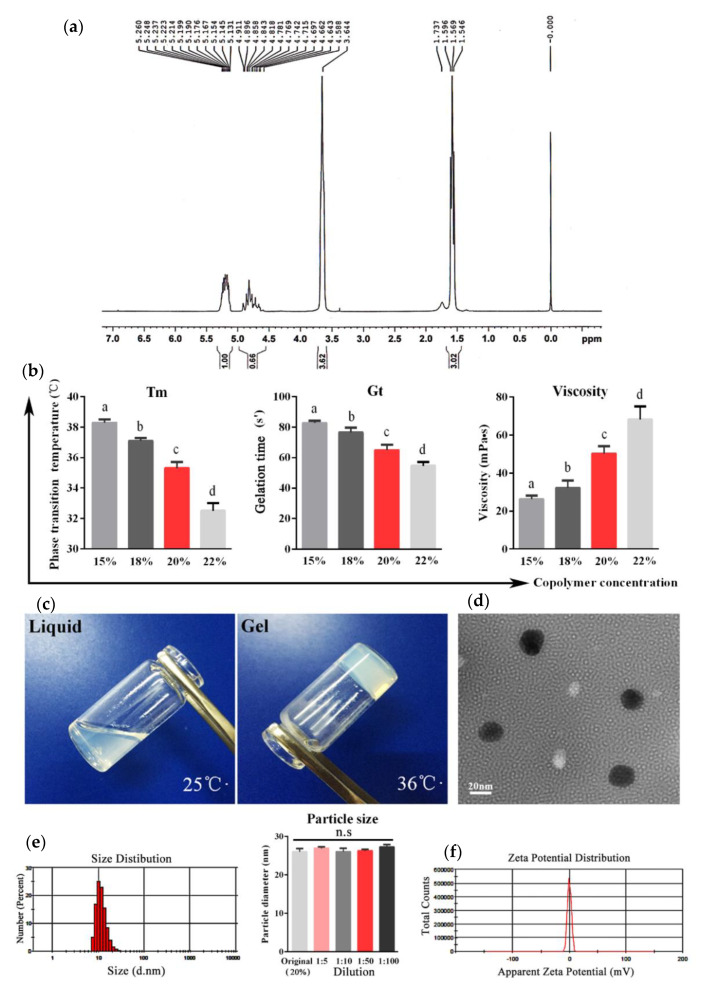
Characterization of the PLGA-PEG-PLGA hydrogel. (**a**) The ^1^H-NMR spectrum of the PLGA-PEG-PLGA triblock copolymer (300 MHz, CDCl_3_). (**b**) The phase transition temperature (T_m_), gelation time (Gt), and viscosity of copolymer solutions with different concentrations. (**c**) Appearance of the copolymer with a concentration of 20% at different temperatures. (**d**) Transmission electron microscopy of the copolymer solution. (**e**) Particle sizes of the original solution and diluent with different dilution ratios of the copolymer based on dynamic light scattering (DLS) analyses. (**f**) Zeta potential of the copolymer micelles calculated by the electrophoretic velocity according to the Henry equation. The results are expressed as the mean ± SD of three independent experiments. Bars with different superscripts mean significant differences (*p* < 0.05).

**Figure 2 molecules-25-02505-f002:**
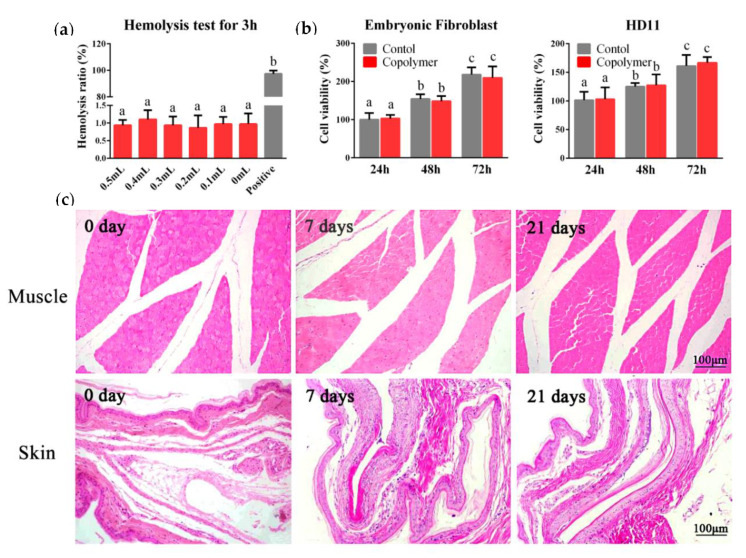
Safety of the PLGA-PEG-PLGA hydrogel. (**a**) The blood compatibility of the material was studied by a hemolysis test. We cultured the chicken blood with 0.5, 0.4, 0.3, 0.2, 0.1, and 0 mL polymer solution (original concentration was 20%) for 3 h, and distilled water was chosen for the positive control to induce hemolysis. (**b**) The cytotoxic test results in the chicken embryonic fibroblast and HD11. (**c**) After intramuscular or subcutaneous injection, the muscle and skin from the injection site were collected for the hematoxylin-eosin (HE) staining to observe the inflammatory cell infiltration. The results are expressed as the mean ± SD of three independent experiments. Bars with different superscripts mean significant differences (*p* < 0.05).

**Figure 3 molecules-25-02505-f003:**
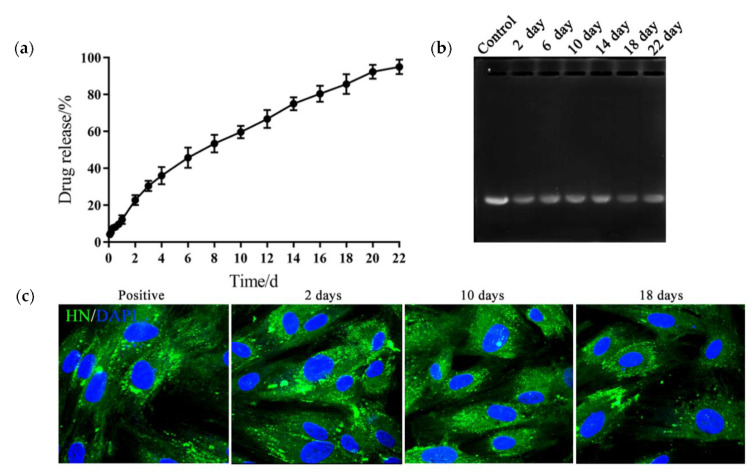
PLGA-PEG-PLGA hydrogel consistently released the DNA vaccine and maintained its biological activity. (**a**) The release curve of the DNA vaccine from the hydrogel. (**b**) The electrophoresis image of the release media. Control: plasmids of the NDV HN gene (pVAX1-HN) plasmids. (**c**) Immune fluorescence images of the HD11 transfected with release media. The release media samples from the hydrogel DNA vaccine were collected at 2, 10, and 18 days, then transfected into HD11 cells with liposomes. After 2 days, the cells were incubated with Newcastle disease virus (NDV)-positive serum and stained with the fluorescein isothiocyanate labelled anti-chicken immunoglobulin G antibody (IgG-FITC), then detected by the confocal microscope. pVAX1-HN plasmids were chosen as the positive control. The results are expressed as the mean ± SD of three independent experiments.

**Figure 4 molecules-25-02505-f004:**
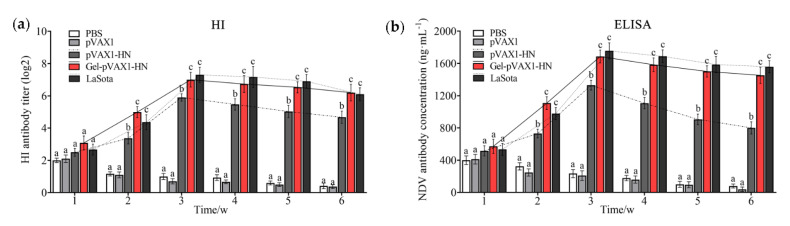
Humoral immune response assay of the NDV DNA hydrogel vaccine. The chickens were inoculated with PBS, pVAX1 plasmids, pVAX1-HN plasmids, and pVAX1-HN plasmids encapsulated in the hydrogel (Gel-pVAX1-HN) and commercial attenuated vaccine (LaSota), then the serum samples were collected at each predetermined time for the hemagglutination inhibition test (**a**) and ELISA assay (**b**). The results are expressed as the mean ± SD of three independent experiments. Bars at the same week with different superscripts mean significant differences (*p* < 0.05).

**Figure 5 molecules-25-02505-f005:**
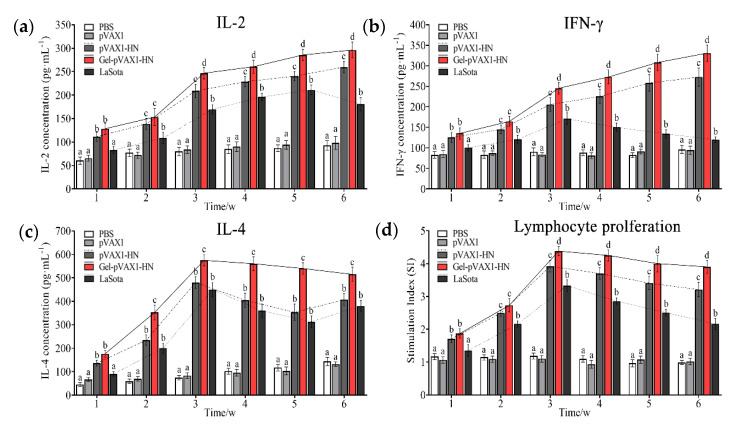
Cellular immune response assay of the NDV DNA hydrogel vaccine. After inoculation, the blood samples were collected at each time point for the next assay. ELISA assay detected the serum levels of IL-2 (**a**), INF-γ (**b**), and IL-4 (**c**). (**d**) Ratios of lymphocyte proliferation in immunized chickens. The results are expressed as the mean ± SD of three independent experiments. Bars at the same week with different superscripts mean significant differences (*p* < 0.05).

**Figure 6 molecules-25-02505-f006:**
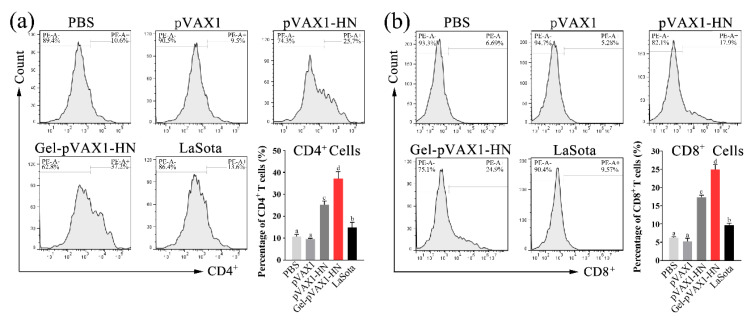
The percentages of CD4^+^ T cells (**a**) and CD8^+^ T cells (**b**) of peripheral blood lymphocytes. At the third week after immunization, the peripheral blood lymphocytes were isolated and stained with the CD4-PE or CD8-PE antibodies then counted by flow cytometry. The results are expressed as the mean ± SD of three independent experiments. Bars with different superscripts mean significant differences (*p* < 0.05).

**Figure 7 molecules-25-02505-f007:**
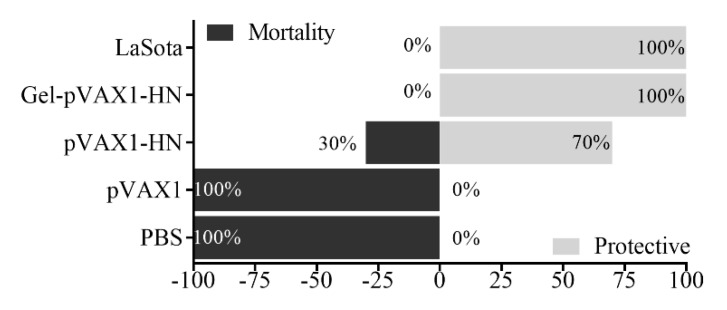
Protective efficacy of the immunized chickens against the highly virulent NDV strain. Chickens in all groups (n = 10) were challenged with the highly virulent NDV strain; after 14 days, the protection and mortality rates were recorded.

**Figure 8 molecules-25-02505-f008:**
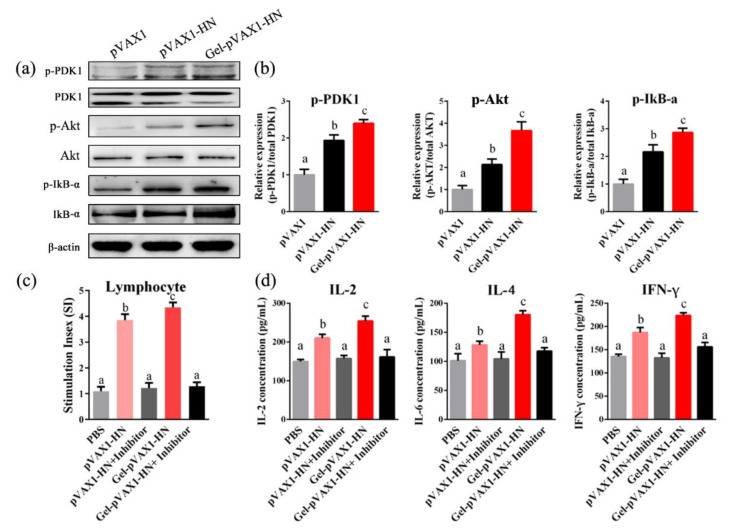
PLGA-PEG-PLGA vaccine delivery system activated the PI3K/Akt pathway in lymphocytes. (**a**) Lymphocytes isolated from the immune chickens’ blood and the protein levels of the PI3K/Akt pathway were detected by a western blot analysis. (**b**) Relative protein expressions. The effects of the PI3K inhibitor on the lymphocyte proliferation (**c**) and cytokine production (**d**). Lymphocytes from the pVAX1-HN and Gel-pVAX1-HN groups were cultured with or without LY294002 (1 μM) for 48 h, and the lymphocyte proliferation abilities and the concentrations of immune cytokines were compared. The results are expressed as the mean ± SD of three independent experiments. Bars with different superscripts mean significant differences (*p* < 0.05).

**Table 1 molecules-25-02505-t001:** The ^1^H-NMR spectrum data analysis of the poly(lactide co-glycolide acid) and polyethylene glycol triblock copolymer (PLGA-PEG-PLGA).

NO.	Chemical Shift (ppm)	Corresponding Proton
1	5.2	DL-lactide CH
2	4.8	Glycolide CH_2_
3	3.6	CH_2_ of PEG
4	1.5	DL-lactide CH_3_

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
