# Peer review of "Development of PLGA-PEG-PLGA Hydrogel Delivery System for Enhanced Immunoreaction and Efficacy of Newcastle Disease Virus DNA Vaccine"

_molecules, 2020, doi:10.3390/molecules25112505_

Round 1
Reviewer 1 Report
The study by Gao et al explored the use of PLGA-PEG-PLGA hydrogels for the delivery of DNA vaccines against Newcastle disease (NDV). The authors demonstrated sustained release, high biocompatibility and an enhanced immune activation/protection against highly virulent NDV strain. I suggest minor changes and clarifications to the manuscript before it is accepted as below:
Minor changes:
There are quite a lot of grammatic errors in the manuscript. The authors are requested to make changes to some of these errors by reading thoroughly and justifying where appropriate.
a. Abstract
line 11: continues threatening the poultry to 'continues to threaten poultry'.
line 15: plasmid encapsulated to 'plasmid was encapsulated'
b. Introduction
line 28: which leads high to 'which leads to high'
line 56: which containing..to "which contains'
c. Results
line 85/86: the concentration at.. to 'the concentration of'
line 87: Polymer concentration of 20 % was said to be transparent. However the image (Figure 1c) shows a slightly opaque formulation. Can the authors clarify or change this in the results section.
line 94: was electrical neutrality to 'was electrically neutral'?
Figure 1d: Can the authors insert arrows in the image to show what the micelles are in the TEM image?
Line 165: at different time after immunization to 'at different time points'...
Figure 6: The statistical significance should be indicated with *, **, *** etc depending on the p value instead of using different alphabets. This is widely accepted in the scientific community and easier for readers to understand than how the authors have presented it in their figures. Please comment or change appropriately.
d. Discussion
Line 222/223:ability to be multivalent vaccine..please rephrase this
Line 223: It also can..to "It can also..'
Line 230: However, the researches...to'However, researchon DNA vaccine delivery...
Line 240: ..accumulation of overmuch charges. Please rephrase, this doesnt sound so scientifc.
Line 272: immunogical.. to 'immunological'
Line 281 (and in many places in the manuscript): The authors refer to their formulation as DNA hydrogel vaccine which is not quite an accurate description considering the fact that there are quite a few hydrogels composed entirely of DNA. Please reconsider referring to the formulation as plasmid loaded hydrogel vaccine since the plasmid is loaded in the PLGA-PEG-PLGA hydrogel.
Methods:
Section 4.5: The authors did not clearly explain how the release media was combined with transfection reagent, in what ratio and for how long before changed into media. I assume the cells were not kept in PBS for 2 days. Better clarification is needed.
I find the release method a bit confusing. A 2ml hydrogel system was incubated with 2 ml release buffer. First, the hydrogel has the potential to imbibe the buffer which means there will be a relatively lower volume of buffer on top/incubated with the gel. There will of course be a slower release because of this. Can the authors justify why this method was used and how they guarded against false slow/sustained release using this volume? Moreover the volume of release buffer collected at each time point was not mentioned in the manuscript.
Author Response
Comments and Suggestions for Authors:
Minor changes:
There are quite a lot of grammatic errors in the manuscript. The authors are requested to make changes to some of these errors by reading thoroughly and justifying where appropriate.
Response:
Thank you for your careful review and suggestions. As recommended, we have revised and adapted the statements in the manuscript accordingly. In terms of English language and style, the manuscript has been edited by MDPI's English editing service and further proofread by our native English speaking colleague.
- Abstract
line 11: continues threatening the poultry to 'continues to threaten poultry'.
line 15: plasmid encapsulated to 'plasmid was encapsulated'
Response:
Line 11: The sentence has been rephrased to “The highly contagious Newcastle disease virus (NDV) is a threat to poultry all over the world.” in the line 11 in our revised manuscript.
Line 15: The statement has been revised to “plasmid was encapsulated” in the line 17.
- Introduction
line 28: which leads high to 'which leads to high'
line 56: which containing..to "which contains'
Response:
Line 28: The statement has been revised to “which leads to high” in the line 32.
Line 56: The statement has been revised to “which contains” in the line 62.
- Results
line 85/86: the concentration at.. to 'the concentration of'
line 87: Polymer concentration of 20 % was said to be transparent. However the image (Figure 1c) shows a slightly opaque formulation. Can the authors clarify or change this in the results section.
line 94: was electrical neutrality to 'was electrically neutral'?
Response:
Line 85/86: The statement has been revised to “the concentration of” in the line 93.
Line 87: In the previous version, our description was not very precise. In fact, the polymer of 20 % is not completely transparent. This sentence has been revised to “The polymer solution at a concentration of 20% was liquid at 25 ℃” in the line 94.
Line 94: The statement has been revised to “was electrically neutral” in the line 101.
Figure 1d: Can the authors insert arrows in the image to show what the micelles are in the TEM image?
Line 165: at different time after immunization to 'at different time points'...
Response:
In Figure 1d, the arrows have been added to show the micelles in the TEM image and indicated in the figure legend.
Line 165: The statement has been revised to “at different time points” in the line 178.
Figure 6: The statistical significance should be indicated with *, **, *** etc depending on the p value instead of using different alphabets. This is widely accepted in the scientific community and easier for readers to understand than how the authors have presented it in their figures. Please comment or change appropriately.
Response:
Thank you very much for your advice. I quite agree with you that the asterisks are the most common symbols for the statistical significance. Nevertheless, in our manuscript which there are many groups and sampling time points, it’s hard to use asterisks to indicate significant differences between groups. If we draw horizontal line between groups, the image would be more complicated. As far as we know, using different alphabets is another common representation form for the statistical method of multiple comparison (Yue Yang, Ronge Xing, Song Liu, et al. 2020. Chitosan, hydroxypropyltrimethyl ammonium chloride chitosan and sulfated chitosan nanoparticles as adjuvants for inactivated Newcastle disease vaccine. Carbohydrate Polymers, 229, 9, https://doi.org/10.1016/j.carbpol.2019.115423). Thanks again for your suggestion. In future studies, we will adopt the form of asterisks to indicate the significance as far as possible.
- Discussion
Line 222/223:ability to be multivalent vaccine..please rephrase this
Line 223: It also can..to "It can also..'
Line 230: However, the researches...to'However, research on DNA vaccine delivery...
Line 240: ..accumulation of overmuch charges. Please rephrase, this doesnt sound so scientifc.
Line 272: immunogical.. to 'immunological'
Response:
Line 222/223: The sentence has been rephrased to “DNA vaccines have several advantages over conventional vaccines, such as security and convenience. DNA vaccine can be made into a multivalent vaccine that encode a variety of antigenic genes.” in the line 239.
Line 223: The statement has been revised to “It can also” in the line 240.
Line 230: The statement has been revised to “However, research on its use as a DNA vaccine delivery system” in the line 248.
Line 240: The sentence has been rephrased to “Polymers with an excessive zeta potential have strong cytotoxicity, because high charges on the cell membrane surface can cause cells to rupture.” in the line 257.
Line 272: The word has been revised to “immunological” in the line 292.
Line 281 (and in many places in the manuscript): The authors refer to their formulation as DNA hydrogel vaccine which is not quite an accurate description considering the fact that there are quite a few hydrogels composed entirely of DNA. Please reconsider referring to the formulation as plasmid loaded hydrogel vaccine since the plasmid is loaded in the PLGA-PEG-PLGA hydrogel.
Response:
Thank you very much for pointing this out. That is true, it should be more accurate to call the formulation in our manuscript as plasmid loaded hydrogel vaccine. Given that in a broad sense, the term “DNA vaccine” involves the direct introduction into appropriate tissues of a plasmid containing the DNA sequence encoding the antigen(s) against which an immune response is sought, and relies on the in situ production of the target antigen (WHO web site https://www.who.int/biologicals/areas/vaccines/dna/en/). DNA vaccines are not usually distinguished from the vaccines that made up of plasmids in the literatures (Kutzler, M. A., & Weiner, D. B. 2008. DNA vaccines: ready for prime time? Nature reviews. Genetics, 9, 776–788. https://doi.org/10.1038/nrg2432). For the sake of consistency, we have revised the “DNA hydrogel vaccine” to “DNA loaded hydrogel vaccine” throughout the revised manuscript.
Methods:
Section 4.5: The authors did not clearly explain how the release media was combined with transfection reagent, in what ratio and for how long before changed into media. I assume the cells were not kept in PBS for 2 days. Better clarification is needed.
Response:
We have added the following description about the transfection operation in section 4.5 in the revised version. “250 μL of release media samples collected at 2, 10, and 18 days was taken as Solution A, respectively. 12 μL of LipoFiter liposome transfection reagent (Hanbio, Shanghai, China) was mixed with DMEM to the final volume of 250 μL (Solution B). Solution A and Solution B was mixed and incubated for 20 min at room temperature. Then the mixture was evenly dropped into the same well and transfected into HD11 cells. After culturing the cells for 6 h at 37 °C, all the culture medium was discarded and fresh DMEM medium was added to each well to continue the cell culture.”
I find the release method a bit confusing. A 2ml hydrogel system was incubated with 2 ml release buffer. First, the hydrogel has the potential to imbibe the buffer which means there will be a relatively lower volume of buffer on top/incubated with the gel. There will of course be a slower release because of this. Can the authors justify why this method was used and how they guarded against false slow/sustained release using this volume? Moreover the volume of release buffer collected at each time point was not mentioned in the manuscript.
Response:
Thank you for pointing this out. The membrane-free dissolution method is commonly used for the in vitro drug release assay from the hydrogel in other literatures (Wang, X., Zhang, Y., Xue, W., et al. 2017. Thermo-sensitive hydrogel PLGA-PEG-PLGA as a vaccine delivery system for intramuscular immunization. Journal of Biomaterials Applications, 31: 923-932. https://doi.org/10.1177/0885328216680343). Based on the above consideration, this method was used to evaluate the release in vitro in our paper. We think your comment is quite reasonable, and this method does have the defects as you mentioned. Nevertheless, in our previous studies, the effects of different volume of release buffer on gel dissolution nd drug release had been examined. The relevant data and results are not shown in this paper. The results showed that the effects did not affect the overall conclusion in the current study, and the polymer does have the drug slow-release effect. In future studies, we will take your advice by improving the method to reduce the impact of this defect. As for the volume of realse buffer collected at each time point, the sentence was revised to “All the release medium was collected at each sampling time point and replaced with fresh buffer of 2 mL, in order to maintain sink conditions.”
Reviewer 2 Report
1. The manuscript is riddled with grammatical errors. Proofreading and editing required to improve the grammar and the structure.
2. In the abstract, HN must be expanded (no context to understand what hemagglutinin-neuraminidase is and why it is a part of the vaccine).
3. In the introduction section (pg2, para2), controlled biodegradation rate is mentioned as an advantage of the polymer. However, the degradation of the polymer inside or outside the biological system is not addressed in the paper. How long does the gel remain in the body/outside the body? How is it degraded?
4. In section 2.2, in vivo safety test is mentioned. The procedure could be included in the methods section.
5. In the same section, it is mentioned that the chicken survived after 14 days. The HE staining figure shows results for day 21. Could be clearer about the timeline. Why 14/21 days?
6. In section 2.4, it is said that the serum samples were collected at predetermined time points. The time points could me more clearly written (week 1, week 2, etc.) to corelate easily with the graphs.
7. In figure 8, the figure legend needs to be corrected.
8. In section 4.1 (or anywhere in the paper), source of HD11 cells and chicken embryonic fibroblast cells are not mentioned.
9. In section 4.3 (methods), the gelation time must be explained clearly (vague about the time period, difficult to understand what the start and end points considered are).
10. In section 4.4, the protocol of the cytotoxicity assay using CCK 8 is wrong (CCK 8 solution is added first and the sample is incubated, then the absorbance is measured). Needs to be corrected. The absorbance wavelength can be mentioned.
11. In section 4.6, the number of animals selected from each group must be written clearly (presently misleading).
Author Response
Comments and Suggestions for Authors:
- The manuscript is riddled with grammatical errors. Proofreading and editing required to improve the grammar and the structure.
Response:
Thank you very much for your careful review and suggestions. As recommended, the manuscript has been edited by MDPI's English editing service and further proofread by our native English speaking colleague.
- In the abstract, HN must be expanded (no context to understand what hemagglutinin-neuraminidase is and why it is a part of the vaccine).
Response:
The following introduction of HN protein has been added in the abstract. “The hemagglutinin-neuraminidase (HN) protein is the main protective antigen of NDV, and has good immunogenicity.”
- In the introduction section (pg2, para2), controlled biodegradation rate is mentioned as an advantage of the polymer. However, the degradation of the polymer inside or outside the biological system is not addressed in the paper. How long does the gel remain in the body/outside the body? How is it degraded?
Response:
Thank you for pointing this out. Actually, the degradation of the polymer in vitro and in vivo had been determined in our previous studies. The relevant data and results are not shown in this paper. Based on the results, the hydrogel of 2 mL could maintain the gel state for 32 days at 37 °C or 28 days at 42 °C in vitro when it was in contact with the same volume of aqueous medium. In the chicken, the hydrogel remained for 21 days after subcutaneous injection of 1 mL, while it maintained the gel form for 24 days after intramuscular injection of 1 mL. When in contact with aqueous medium, the polymer would conduct hydrolysis reactions, and the ester bonds between the blocks would be gradually disconnected, resulting in the copolymer degradation into small fragments.
- In section 2.2, in vivo safety test is mentioned. The procedure could be included in the methods section.
Response:
The procedure of in vivo safety test has been added in the methods section (Section 4.4) in the revised version. “14-day-old chickens were randomly divided into 2 groups (15 chickens per group). Each chicken was inoculated with the polymer solution of 1 mL by subcutaneous or intramuscular injection, respectively, and was observed continuously for 14 days to evaluate the clinical symptoms and toxic reactions.”
- In the same section, it is mentioned that the chicken survived after 14 days. The HE staining figure shows results for day 21. Could be clearer about the timeline. Why 14/21 days?
Response:
According to the Technical Guidelines For Drug Research issued by the National Medical Products Administration of China (http://www.nmpa.gov.cn/WS04/CL2182/299975.html), the procedure of in vivo acute toxicity test is described as follows. “In the acute toxicity test, it is required to continuously observe the toxic reaction and death of animals for 7~14d after a single double dose or multiple doses of the drug in one day.” Furthermore, the similar approaches of safety test are often used in other literatures (Yue Yang, Ronge Xing, Song Liu, et al. 2020. Chitosan, hydroxypropyltrimethyl ammonium chloride chitosan and sulfated chitosan nanoparticles as adjuvants for inactivated Newcastle disease vaccine. Carbohydrate Polymers, 229, 9, https://doi.org/10.1016/j.carbpol.2019.115423). Based on the above considerations, the immunized chickens were continuously observed for 14 days.
Besides, the following statements have been added in section 4.4 to definite the timeline. “Results of our previous studies indicated that the hydrogel could remain for 21 or 24 days after subcutaneous or intramuscular injection of 1 mL, respectively. For consistency of the sampling time points, the skin or muscle samples from the injection sites were collected for HE staining at 0, 7, and 21 days to observe the hydrogel biocompatibility.”
- In section 2.4, it is said that the serum samples were collected at predetermined time points. The time points could me more clearly written (week 1, week 2, etc.) to corelate easily with the graphs.
Response:
In section 2.4, the description about the time points that the serum samples were collected has been revised to “the serum samples were collected at week 1, 2, 3, 4, 5 and 6 after immunization.”
- In figure 8, the figure legend needs to be corrected.
Response:
In figure 8, the caption has been rephrased to “Figure 8. The PLGA-PEG-PLGA vaccine delivery system enhanced the activation of the PI3K/Akt pathway in the lymphocytes.” Besides, some modifications have been made in the figure legend.
- In section 4.1 (or anywhere in the paper), source of HD11 cells and chicken embryonic fibroblast cells are not mentioned.
Response:
The source of HD11 cells and chicken embryonic fibroblast cells has been replenished in section 4.1. “The highly virulent NDV (F48E9 strain), the NDV positive serum and the chicken macrophages (HD11) were obtained from the Jiangsu Academy of Agricultural Sciences (Nanjing, China). Chicken embryo fibroblasts (CEF) were isolated from 9-day-old SPF chicken embryos that were purchased from QYH Biotech Co. Ltd (Nanjing, China).”
- In section 4.3 (methods), the gelation time must be explained clearly (vague about the time period, difficult to understand what the start and end points considered are).
Response:
The following description about the determination of the gelation time has been added in section 4.3. “In the process of continuous warming, timing was started when the solution temperature reached 25 °C, suspended when observing the fluidity of the content with the tilt of the tube and ended at the moment when the solution had been converted to a hydrogel. This period was defined as the gelation time (Gt).”
- In section 4.4, the protocol of the cytotoxicity assay using CCK 8 is wrong (CCK 8 solution is added first and the sample is incubated, then the absorbance is measured). Needs to be corrected. The absorbance wavelength can be mentioned.
Response:
Thank you for pointing this out. In the previous version, the statement was brief. The following description has been replenished in section 4.4 in the revised manuscript. “After culturing the cells on the plates overnight, the polymer solution was added and co-cultured for 20 h, 44 h or 68 h at 37 °C. Then, 10 μL of CCK-8 solution was added into the medium, and culturing of the cells continued for 4 h. Finally, the absorbance at 450 nm was detected.”
- In section 4.6, the number of animals selected from each group must be written clearly (presently misleading).
Response:
The number of animals selected from each group has been added in section 4.6. “(20 chickens per group)”
Reviewer 3 Report
nice manuscript.
Avoid active sentences in abstract. significance of the work should be more focused.
Author Response
Comments and Suggestions for Authors:
nice manuscript. Avoid active sentences in abstract. significance of the work should be more focused.
Response:
Thank you very much for your comments and suggestions. As recommended, some modifications have been made in the abstract to avoid active statement as far as possible. To highlight the significance of the work, the following statement has been replenished in the abstract. “It is helpful to improve the practical application of NDV DNA vaccine in the poultry industry, to effectively prevent and control the occurrence of ND.” And the sentences have been rephrased at the end of the discussion. “The results of the present study indicate that this hydrogel vaccine was able to enhance humoral immunity, cellular immunity and the antiviral ability of the body with longer immunity protection. It is proved that the PLGA-PEG-PLGA thermosensitive hydrogel can be used as a promising DNA vaccine delivery system, to induce potent immunoreactions and increase vaccine potency. Furthermore, the hydrogel can be used as carrier with safety and efficiency for other types of vaccines and has increasing potential in practical use.”
Reviewer 4 Report
The topic of this manuscript (NDV vaccines) is of great interest. The use of hydrogel for the controlled release, even if not so novel, has great potential even in practical use. The manuscript is well written and all the results are fully supported by exeprimental data. Results are clear and all the methods are adequatley described. In my opinion this manuscript should be accepted for publication in the present form
Author Response
Comments and Suggestions for Authors:
The topic of this manuscript (NDV vaccines) is of great interest. The use of hydrogel for the controlled release, even if not so novel, has great potential even in practical use. The manuscript is well written and all the results are fully supported by exeprimental data. Results are clear and all the methods are adequatley described. In my opinion this manuscript should be accepted for publication in the present form
Response:
Thank you very much for your comments and suggestions. The results of the present study indicate that the PLGA-PEG-PLGA thermosensitive hydrogel can be used as a promising DNA vaccine delivery system, to induce potent immunoreactions and increase vaccine potency. Furthermore, the hydrogel can be used as carrier with safety and efficiency for other types of vaccines and has increasing potential in practical use. For better effects of controlled release, we’re thinking about using the hydrogel delivery system in combination with other drug delivery systems in future studies, such as magnetic nanoparticles or photosensitive materials, to establish a multi-response drug delivery system for the biological macromolecular (e.g., proteins, peptides) and other treatment components (e.g., vaccine, antibody) for improving the kinetics of drug release. Thanks again for your advice. We will pay more attention to carry out more innovative work in future.
Round 2
Reviewer 2 Report
The authors have made all the necessary changes as suggested earlier.